# Fungal effector Ecp6 outcompetes host immune receptor for chitin binding through intrachain LysM dimerization

Andrea Sánchez-Vallet[1,2†], Raspudin Saleem-Batcha[3†], Anja Kombrink[2], Guido Hansen[3], Dirk-Jan Valkenburg[2], Bart PHJ Thomma[2,4*§], Jeroen R Mesters[3*§]

[1]Centro de Biotecnología y Genómica de Plantas, Universidad Politécnica de Madrid, Madrid, Spain; [2]Laboratory of Phytopathology, Wageningen University, Wageningen, Netherlands; [3]Institute of Biochemistry, Center for Structural and Cell Biology in Medicine, University of Lübeck, Lübeck, Germany; [4]Centre for BioSystems Genomics, Wageningen, Netherlands

**Abstract** While host immune receptors detect pathogen-associated molecular patterns to activate immunity, pathogens attempt to deregulate host immunity through secreted effectors. Fungi employ LysM effectors to prevent recognition of cell wall-derived chitin by host immune receptors, although the mechanism to compete for chitin binding remained unclear. Structural analysis of the LysM effector Ecp6 of the fungal tomato pathogen *Cladosporium fulvum* reveals a novel mechanism for chitin binding, mediated by intrachain LysM dimerization, leading to a chitin-binding groove that is deeply buried in the effector protein. This composite binding site involves two of the three LysMs of Ecp6 and mediates chitin binding with ultra-high (pM) affinity. Intriguingly, the remaining singular LysM domain of Ecp6 binds chitin with low micromolar affinity but can nevertheless still perturb chitin-triggered immunity. Conceivably, the perturbation by this LysM domain is not established through chitin sequestration but possibly through interference with the host immune receptor complex.

**\*For correspondence:** bart. thomma@wur.nl (BPHJT); mesters@biochem.uni-luebeck. de (JRM)

†These authors contributed equally to this work

§These authors also contributed equally to this work

**Competing interests:** The authors declare that no competing interests exist.

## Introduction

Fungi constitute an evolutionarily and ecologically diverse group of microorganisms. Although most species are saprophytic, many are causative agents of disease and include plant pathogens that cause considerable yield losses in agricultural crops worldwide (*Skamnioti and Gurr, 2009*; *Oliver, 2012*). To sense the presence of potential pathogens, hosts employ cell surface receptors that detect conserved pathogen-associated molecular patterns (PAMPs) to activate immunity (*Medzhitov and Janeway, 1997*; *Boller and Felix, 2009*). Chitin, an *N*-acetyl-D-glucosamine (GlcNAc) homopolymer, is the primary structural component of fungal cell walls and is recognized as a PAMP by plant cell surface receptors that contain extracellular lysin motifs (LysMs) (*Felix et al., 1993*; *Shibuya et al., 1993*; *Shibuya et al., 1996*; *Kombrink et al., 2011*). The first chitin receptor has been cloned from rice (*Oryza sativa*) as the chitin oligosaccharide elicitor-binding protein (CEBiP; *Kaku et al., 2006*) that forms a receptor complex in a ligand-dependent manner that furthermore includes the chitin elicitor receptor kinase-1 (OsCERK1; *Shimizu et al., 2010*). Similarly, *Arabidopsis thaliana* CERK1 binds chitin and is required for chitin-triggered immunity (*Miya et al., 2007*; *Wan et al., 2008*; *Iizasa et al., 2010*; *Petutschnig et al., 2010*). However, no homologs of rice CEBiP could be implicated in chitin-triggered immunity against fungal infection in *Arabidopsis* (*Shinya et al., 2012*; *Wan et al., 2012*). Recently, a crystal structure of the AtCERK1 ectodomain was determined, revealing chitin binding to one of the three AtCERK1 LysMs only (Protein Data Bank code 4EBZ; *Liu et al., 2012*). LysM2 of AtCERK1 binds three GlcNAc residues of a longer chitin oligomer in a shallow groove on the surface of the protein, with both

**eLife digest** The ability to launch an immune response is not unique to animals. Plants have also evolved the ability to detect molecules present on the surface of pathogens such as fungi. These molecular signatures are known as pathogen-associated molecular patterns (PAMPs), and they are detected by specialized receptors on the surface of plant cells.

Chitin, the main structural component of the cell wall in fungi, is one example of a PAMP. Many species of plants are able to detect chitin using receptors that contain sequences of amino acids called lysin motifs. Previous work in the model plant Arabidopsis has shown that chitin binds to a single lysin motif within each plant receptor.

However, just as plants have evolved the ability to recognize PAMPs, so fungi have evolved ways to outwit plants. They have developed small molecules called effector proteins that bind to PAMPs, in effect hiding them from the plant receptors. The tomato fungus *Cladosporium fulvum*, for example, secretes an effector protein called Ecp6, which contains lysin motifs just like those in the plant receptors. By binding chitin fragments, Ecp6 helps the fungus to avoid detection by its host plant.

Now, Sánchez-Vallet et al. present the high resolution crystal structure of Ecp6 and reveal the mechanism by which it outcompetes the plant's own chitin receptors. In the presence of chitin, two lysin binding motifs within the Ecp6 protein combine to produce a binding site with ultrahigh affinity for chitin. This can outcompete the plant receptors, which use only a single lysin domain to bind the fungal protein.

As well as providing a molecular explanation for how certain fungi manage to evade the immune response in plants, the work of Sánchez-Vallet et al. offers an unusual example of convergent evolution, in which two evolutionarily distant organisms have evolved the ability to recognize the same molecule through structurally diverse proteins.

ends of the oligomer as well as one half side of each of the three bound GlcNAc residues protruding into the solvent (*Liu et al., 2012*). Biochemical experiments suggest that sufficiently long chitin oligomers act as bivalent ligands, leading to ligand-induced AtCERK1 dimerization that is required for immune signaling (*Liu et al., 2012*).

The evolution of interactions between microbial pathogens and their hosts involves a continuous arms race, in which pathogens secrete effectors to deregulate host immunity (*de Jonge et al., 2011*). The leaf mold fungus *Cladosporium fulvum* abundantly secretes the LysM-containing effector protein Ecp6 (for extracellular protein 6; *Bolton et al., 2008*) during colonization of its host tomato. Ecp6 acts as a scavenger of chitin fragments and thus prevents recognition of the fungus by host immune receptors (*de Jonge et al., 2010*). Intriguingly, Ecp6 homologs occur throughout the fungal kingdom, suggesting a fundamental role of chitin scavenging in fungal pathogenicity (*Bolton et al., 2008*; *de Jonge and Thomma, 2009*). Indeed, LysM effectors from the fungal wheat pathogen *Mycosphaerella graminicola* and the rice pathogen *Magnaporthe oryzae* suppress chitin-triggered immunity (*Marshall et al., 2011*; *Mentlak et al., 2012*). However, the mechanism by which LysM effectors outcompete plant receptors for chitin binding remain unknown thus far.

LysM-containing proteins are broadly distributed in bacteria, plants, fungi, and animals (*Buist et al., 2008*; *Kombrink et al., 2011*). Nevertheless, only few tertiary LysM structures have been reported in addition to that of AtCERK1 (*Bateman and Bycroft, 2000*; *Bielnicki et al., 2006*; *Koharudin et al., 2011*; *Liu et al., 2012*). The canonical three-dimensional LysM domain structure consists of a βααβ-fold, in which two α-helices are packed against one side of a two-stranded antiparallel β-sheet. Based on NMR spectroscopy, the loop between the first β-sheet and the first α-helix and the loop between the second α-helix and the second β-sheet were shown to physically interact with chitin oligomers in a 1:1 stoichiometry with an affinity binding constant of up to 21 μM (*Ohnuma et al., 2008*; *Koharudin et al., 2011*). The AtCERK1 ectodomain binds chitin oligomers with a similar affinity, up to 45 μM for $(GlcNAc)_8$, as determined by isothermal titration calorimetry (*Liu et al., 2012*).

Here, we report the crystal structure of Ecp6 to a resolution of 1.6 Å, revealing a novel mechanism for chitin binding by LysMs that developed in fungi, involving substrate-induced intrachain dimerization of two LysM domains to form a buried intramolecular chitin-binding groove. The composite LysM1–LysM3 binding site shows ultra-high chitin-binding affinity, thus explaining how LysM effectors outcompete plant host receptors for chitin binding.

## Results

### Ecp6 crystal structure reveals ligand-induced intrachain LysM dimerization

To understand the molecular mechanism of LysM effector chitin scavenging, we pursued a crystal structure of the *C. fulvum* LysM effector Ecp6. To this end, Ecp6 was heterologously produced in the yeast *Pichia pastoris* and purified. Initial Ecp6 vapor-diffusion crystallization screening consistently yielded intertwined plate-like crystals that grew overnight. These crystals were harvested, smashed in stabilizing mother liquid and used for micro-seeding (*Bergfors, 2003*). In this manner, large Ecp6 protein crystals that belonged to space group $P3_221$ were obtained. The structure was determined by single-wavelength anomalous dispersion (SAD) and refined to a resolution of 1.6 Å with an $R_{work}$ and $R_{free}$ of 20.3% and 22.5%, respectively (*Figure 1* and *Table 1*). The structure model comprises nearly the complete mature protein sequence, from amino acid residues 7 to 195. Residues 1–6 and 196–199 of the structure were not defined in the electron density map, suggesting that they occur as flexible tails on both ends of the protein.

Ecp6 has a tightly packed structure that is stabilized by four disulfide bridges, and two spatially close glycosylation sites (Asn104 and Asn193) were identified. The three LysMs of Ecp6 (LysM1–LysM3) share the typical βααβ-fold, with LysM1 being separated by a long and flexible linker from a compact and rigid body formed by LysM2 and LysM3 (*Figure 1A*). Furthermore, the crystal packing revealed the existence of an Ecp6 homodimer that was also observed in gel filtration chromatography during protein purification, with a flat buried surface of 943 Å² at LysM2 and LysM3 as calculated using PISA (Protein Interfaces, Surfaces and Assemblies; http://www.ebi.ac.uk/pdbe/prot_int/pistart.html; *Figure 1A*; *Krissinel and Henrick, 2007*).

Unexpectedly, the calculated $2F_o-F_c$ map showed a well-defined electron density for a bound chitin tetramer (GlcNAc)$_4$ in a large interdomain groove between LysM1 and LysM3 (~16 Å long and ~11 Å across its widest points; *Figure 1B*), although the protein crystallized in the absence of exogenously added chitin. The chitin oligomer was likely co-purified with Ecp6 and derived from the cell wall of the *P. pastoris* expression system. The observation that the chitin oligomer remained adhered to Ecp6 during the protein purification procedure suggests that it is bound to the groove with high affinity. Thus, our finding illustrates the remarkable potential of Ecp6 to instantly and strongly bind chitin after secretion by fungal cells, a scavenging activity that is expected to similarly occur after secretion by *C. fulvum* to prevent chitin oligosaccharides from activating host immune receptors (*de Jonge et al., 2010*). The loop between the first β-sheet and the first α-helix and the loop between the second α-helix and the second β-sheet in both LysM1 and LysM3 interact with chitin (*Figure 1*), similar to the previously reported chitin binding by single LysM domains including LysM2 of AtCERK1 (*Ohnuma et al., 2008*; *Koharudin et al., 2011*; *Liu et al., 2012*).

Evidently, LysM1 and LysM3 of Ecp6 cooperate to bring two chitin-binding regions together, composing a novel type of binding groove in which one chitin tetramer is nearly completely buried and engaged in many noncovalent interactions, including 12 hydrogen bonds (*Figure 1C*). Specifically, the groove is shaped by the amino acids [20]GDTLT[24] and [46]PNLIEL[51] of LysM1 and [150]GDLFV[154] and [176]PSKL[179] of LysM3. The chitin binding is further strengthened by noncovalent interactions among proximate residues on the complementary surfaces of LysM1 (N[45] and [47]NLIE[50]) and LysM3 ([150]GDLFVD[155]) (*Figure 1*). The concerted action of two LysM domains that compose a single binding groove and the many noncovalent bonds that are involved in the interaction with chitin may collectively provide a mechanistic explanation for high affinity substrate binding, resulting in sequestration by Ecp6 of cell wall chitin from the heterologous host *P. pastoris*. Furthermore, the solute-exposed ends of the chitin tetramer in the structure suggest that longer oligomers can also be bound to the groove, in which case four GlcNAc units of a chitin oligomer make direct contact with the protein while the remaining parts of the molecule protrude into the solvent (*Figure 1*).

### LysM2 of Ecp6 also contains a functional chitin-binding site

To investigate the contribution of the LysM1–LysM3 binding groove to chitin scavenging by Ecp6, mutants in the chitin-binding site of LysM1 and LysM3 were generated (*Figures 1D and 2A*), produced in *P. pastoris*, and tested for the ability to suppress chitin-triggered immunity in tomato cells (*de Jonge et al., 2010*). Two residues that are in close contact with the chitin oligomer and present in the loop between the first β-sheet and the first α-helix (T[22] and L[152]) and in the loop between the second α-helix and the second β-sheet (N[47] and S[177]) of LysM1 and LysM3 were chosen to disrupt chitin binding to the

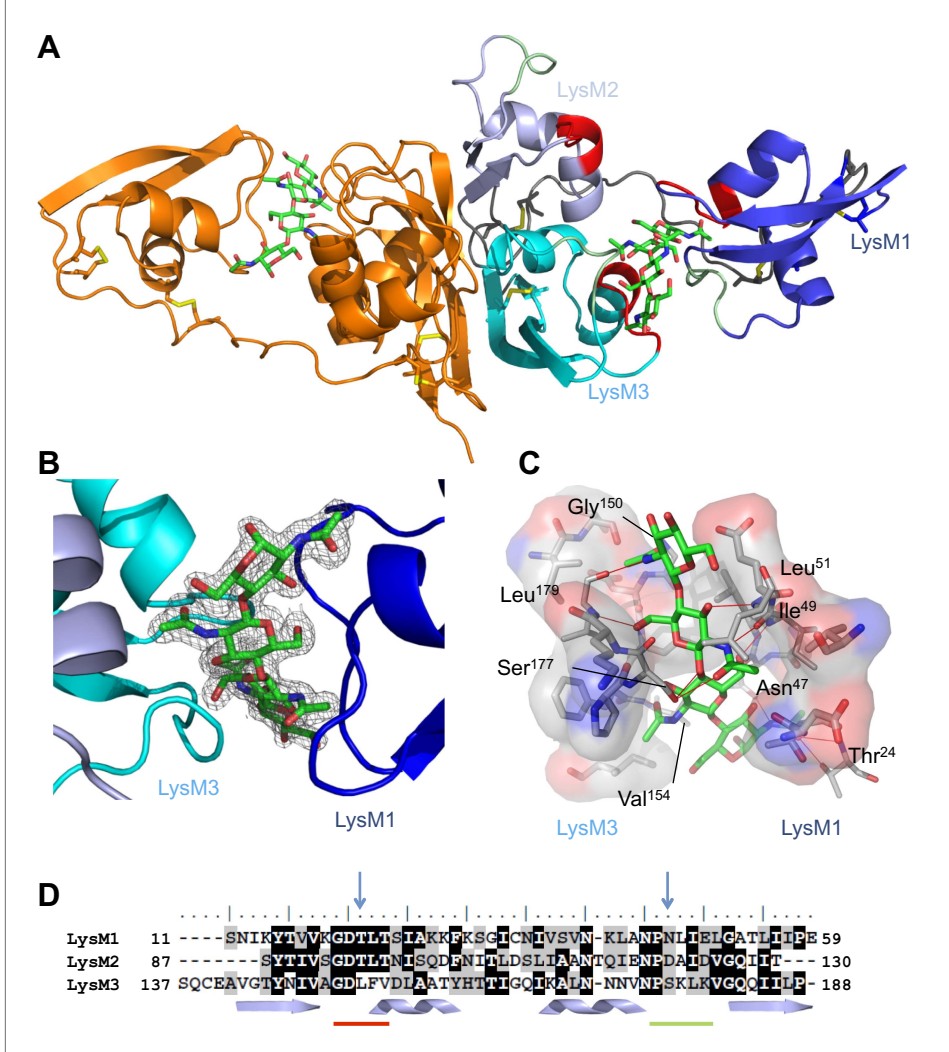

**Figure 1**. Overall crystal structure of the *Cladosporium fulvum* LysM effector Ecp6. (**A**) Crystal structure model of an Ecp6 dimer in which the left monomer is colored orange and the three LysMs of the right monomer are indicated in three shades of blue with the flexible loop between LysM1 and LysM2 in gray. The chitin tetramer (green sticks) and four disulfide bridges (yellow sticks) are indicated. Furthermore, in the right monomer, the two (putative) chitin-binding loops are shown in red and green for each of the LysMs. (**B**) Omit map ($2F_o-F_c$; contoured at 1σ above the mean) with phases calculated omitting (GlcNAc)$_4$. LysMs are colored in three shades of blue as in panel **A**. (**C**) Interactions between Ecp6 and (GlcNAc)$_4$. Hydrogen bonds are indicated in red, and atoms involved in hydrophobic contacts are represented with transparent surface. Only residues forming H-bonds with the chitin are labeled. (**D**) Clustal-W alignment of the three LysM domains of Ecp6. The distribution of the α-helices (helices) and β-sheets (arrows) are shown. The two chitin-binding sites in LysM1 and LysM3 are indicated with a red line for the first loop between the first β-sheet and the first α-helix and a green line for the second loop between the second α-helix and the second β-sheet, as indicated in the right monomer in panel **A**. Blue arrows point towards residues targeted for mutagenesis in the three LysM domains.

LysM1–LysM3 groove (*Figures 1D and 2A*). The selected residues were substituted by relatively large arginine or lysine amino acids to maximize interference with chitin binding. Circular dichroism (CD) spectra of the mutant proteins were obtained to confirm the correct folding.

Isothermal titration calorimetry (ITC) analysis demonstrated that all mutants were able to bind chitin (*Figure 3*). It was previously reported that Ecp6 protein, carrying three LysM domains, binds chitin with 3:1 stoichiometry (*de Jonge et al. 2010*) by building on the general observation that LysM domains bind their substrate with 1:1 stoichiometry. However, the crystal structure revealed that only LysM2 is

**Table 1.** Data collection and refinement statistics

| | Native | SAD |
|---|---|---|
| Data collection statistics | | |
| Beamline | BL14.1 - BESSY | ID29 - ESRF |
| Wavelength (Å) | 0.91814 | 1.70 |
| Space group | P 3$_2$ 2 1 | |
| Cell dimensions a, b, c (Å) | 57.5, 57.5, 118.7 | 57.9, 57.9, 119.7 |
| Resolution (Å) | 49.80–1.59 (1.68–1.59) | 46.24–2.10 (2.21–2.1) |
| $R_{sym}$* (%) | 5.1 (44.9) | 6.7 (40.9) |
| I/σI† | 20.6 (4.4) | 24.2 (2.9) |
| Completeness (%) | 98.4 (94.1) | 97.7 (85.9) |
| Redundancy | 9.2 (8.2) | 15.4 (5.6) |
| Phasing statistics (2.5 Å resolution cut-off) | | |
| Anomalous completeness (%) | – | 96.9 (81.2) |
| Anomalous multiplicity | – | 8.3 (3.0) |
| Figure of Merit (FOM) | – | 0.372 |
| Map Skew | – | 0.14 |
| Correlation of local R.m.s. density | – | 0.82 |
| Correlation Coefficient (CC) | – | 0.76 |
| Refinement statistics | | |
| Resolution (Å) | 49.8–1.6 | 31.0–2.10 |
| No. of reflections (work/free) | 29,078/1546 | 13,072/689 |
| $R_{work}$/$R_{free}$‡ (%) | 20.3 (24.0)/22.5 (27.1) | 21.0 (23.3)/26.6 (34.2) |
| No. of atoms/average B-factor | | |
| Protein | 1392/17.6 | 1387/34.33 |
| Water | 119/37.5 | 53/44.61 |
| Other | 99/24.3 | 115/39.90 |
| R.m.s. deviations bond lengths (Å) | 0.016 | 0.018 |
| R.m.s. deviations bond angles (°) | 1.81 | 1.94 |
| Ramachandran plot (% preferred region/% allowed region) | 96.74/3.26 | 95.72/4.28 |

The values in the parentheses refer to the highest resolution shell.
*$R_{sym}$ = ($\sum |I_{hkl} - <I_{hkl}>|$)/($\sum I_{hkl}$), where the average intensity $<I_{hkl}>$ is taken over all symmetry equivalent measurements and $I_{hkl}$ is the measured intensity for any given reflection.
†I/σI is the mean reflection intensity divided by the estimated error.
‡$R_{work}$ = $||F_o| - |F_c||/|F_o|$, where $F_o$ and $F_c$ are the observed and calculated structure factor amplitudes, respectively. $R_{free}$ is equivalent to $R_{work}$ but calculated for 5% of the reflections chosen at random and omitted from the refinement process.

available for chitin binding when Ecp6 is produced in *P. pastoris*, as the LysM1–LysM3 groove is occupied by chitin (*Figure 1*). Therefore, we integrated the heat measurements using a single binding site model, confirming that *P. pastoris*-produced Ecp6 binds chitin with 1:1 stoichiometry and a dissociation constant (k$_d$) of 4.5 μM (*Figure 3A*). The T22R mutant in LysM1 binds chitin with similar thermodynamic values as Ecp6 produced in *P. pastoris* (k$_d$ = 4.69 μM; n = 0.75; *Figure 3D*). However, chitin binding by the mutants N47K, L152R and S177K did not follow a sigmoidal curve, suggesting that more than one binding event occurs in these mutants (*Figure 3*). Binding by the N47K mutant could be fitted to a 'two binding site' model, revealing that the second binding displayed similar biochemical characteristics (k$_d$ = 5.2 μM; n = 0.984) as Ecp6 produced in *P. pastoris* (*Figure 3E*). These results likely reflect chitin binding to LysM2 and to the partially disrupted LysM1–LysM3 groove, as only one of these two LysMs was mutagenized in a single mutant.

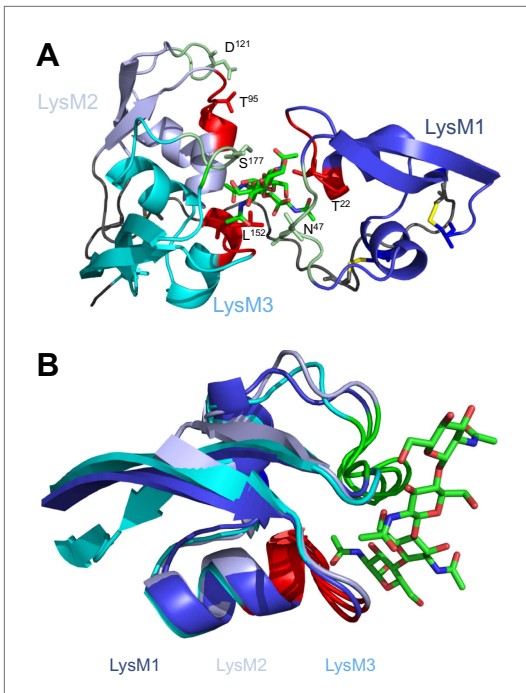

**Figure 2**. Structural and mutational analysis of chitin binding. (**A**) Ecp6 monomer in which the residues targeted for mutagenesis are labeled and represented using sticks. The chitin oligomer is in green sticks. (**B**) Structural alignment of the three Ecp6 LysM domains. Each of the LysMs are colored in three shades of blue and the chitin tetramer is in green sticks. The two chitin-binding loops are shown in red and green for each of the LysMs, as indicated in **Figure 1A**.

Next, we assessed the ability of the various mutants to suppress chitin-triggered immunity in tomato cells. Unfortunately, the two mutants that were generated in LysM1 (T22R and N47K) resulted in an autoactive protein that triggered a response of the tomato cells, leading to medium alkalinization already in the absence of chitin (**Figure 4A**). Most probably the induction of medium pH shift by the mutants in LysM1 was due to improper protein folding, as is suggested by CD spectra when compared with wild-type Ecp6 protein (**Figure 4B**). Possibly, these mutant proteins carry *P. pastoris* chitin with low affinity that is released in the cell suspension, thus activating a pH shift in the absence of exogenously added chitin. Consequently, these mutants could not be assessed for their ability to suppress chitin-triggered immunity. In contrast, the CD spectra of the two mutants in LysM3 (L152R and S177K) were very similar to that of wild-type Ecp6 protein, indicating that they adopt the same secondary structure (**Figure 4D**). As these mutants (L152R and S177K) did not trigger a response of the tomato cells in the absence of chitin oligosaccharides, they could be tested for their scavenging ability. Interestingly, both mutants still prevented chitohexaose [(GlcNAc)$_6$]-induced medium alkalinization. These results suggest that also LysM2 contributes to the suppression of chitin-triggered immunity by *P. pastoris*-produced Ecp6 in the cell suspension assay (**Figure 4C**). Indeed, the high sequence and tertiary structure conservation of the three LysM domains of Ecp6 suggests that LysM2 contains a functional chitin-binding site

(**Figure 1D**; **Figure 2B**). To investigate this, conserved residues in the putative chitin-binding site of LysM2 were selected for mutagenesis (**Figures 1D and 2A**). Based on the structural alignment, T$^{95}$ and D$^{121}$ were mutagenized because of their localization in the two loops that may be involved in chitin binding by LysM2. *P. pastoris*-produced mutant proteins (that contain chitin in the LysM1–LysM3 groove) were tested for their capacity to suppress chitin-triggered immunity (**de Jonge et al., 2010**). Interestingly, mutants T95R and D121K no longer suppressed the chitin oligosaccharide-induced pH shift, revealing that LysM2 contains a functional chitin-binding site that contributes to the suppression of chitin-triggered immunity in the cell suspension assay (**Figure 5A**). Indeed, ITC experiments confirmed that the *P. pastoris*-produced mutants T95R and D121K were impaired in the binding of exogenously added chitohexaose (**Figure 5C and 5D**). Correct folding of the LysM2 mutants was confirmed by CD spectra that were similar to that of wild-type Ecp6 protein (**Figure 5B**). Collectively, these data confirm that suppression of chitin-triggered immunity in the tomato cell suspension by *P. pastoris*-produced Ecp6 is established through the µM chitin-binding activity of LysM2 and explains why chitin scavenging by LysM3 mutants was not impaired. However, calculation of the equilibrium between Ecp6 protein and chitin oligosaccharide levels at the concentrations that were used, and that were well below the measured dissociation constant of LysM2, reveals that only a small amount of the available chitin oligosaccharide is bound by this LysM. Consequently, the suppression of chitin-triggered immunity by LysM2 is unlikely to work via chitin oligosaccharide sequestration.

## The LysM1–LysM3 interdomain groove binds chitin with ultrahigh affinity

Based on the assumption that we have not assessed the full chitin-binding capacity of the Ecp6 effector protein thus far, but only assessed the µM affinity of the solitary LysM2, we attempted to

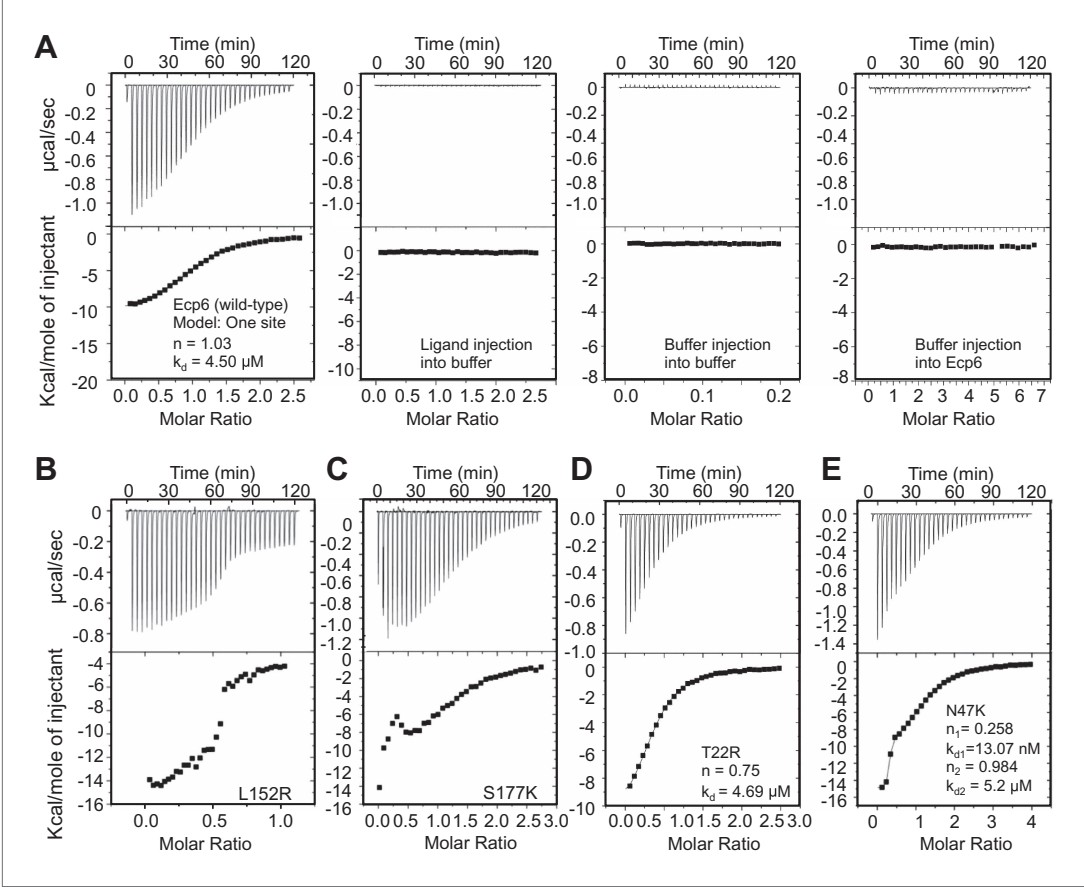

**Figure 3**. Mutants in LysM1 and LysM3 chitin-binding site still bind chitin. Raw data (upper panels) and integrated heat measurements (lower panels) from isothermal titration calorimetry of $(GlcNAc)_6$ binding to Ecp6 produced in *P. pastoris* (**A**) and mutants in LysM3 (**B** and **C**) and in LysM1 (**D** and **E**). Lines in the lower panel represent best-fit curves for one binding site model. ITC control experiments involving chitin ligand into the buffer (PBS), of buffer injection into the buffer, and buffer injection into Ecp6 protein solution are included in panel **A**.

obtain chitin-free Ecp6 protein. Unfortunately, we did not succeed in recovering functional chitin-free Ecp6 after denaturing with 8 M urea. As an alternative strategy, the production of Ecp6 in mammalian (HEK293) cells was pursued. A relatively small amount (4 mg) of chitin-free Ecp6 was obtained from these cells. We confirmed that the Ecp6 protein produced in mammalian cells was able to suppress chitin-triggered immunity in the tomato cell suspension assay (*Figure 6*) and subsequently performed substrate affinity measurements (*Figure 7*). Interestingly, ITC revealed biphasic binding of the chitin hexamer $(GlcNAc)_6$ to chitin-free Ecp6. A first binding phase in which one chitin molecule was bound with ultra-high affinity ($k_d$ = 280 pM; n = 0.99) occurred, followed by binding of an additional molecule with lower affinity ($k_d$ = 1.70 µM; n = 1.03) (*Figure 7B*). Both binding events displayed 1:1 stoichiometry, demonstrating that the three LysM domains of a single Ecp6 monomer are collectively involved in two binding events (*Figure 7B*). As *P. pastoris*-produced Ecp6, in which the LysM1–LysM3 groove is blocked by a *P. pastoris* chitin fragment, displayed similar characteristics as the second binding event of mammalian cell-produced Ecp6 (*Figure 7A,B*), we conclude that the µM affinity should be attributed to LysM2. To confirm this hypothesis, we also produced the LysM2 mutant T95R, which only has the LysM1–LysM3 site available for chitin binding, in mammalian cells. As this mutant binds a single chitin molecule with high affinity ($k_d$ = 4.95 nM) (*Figure 7C*), we concluded that the ultra-high chitin-binding affinity displayed by wild-type Ecp6 corresponds to the LysM1–LysM3 binding groove. The slightly lower dissociation constant that is measured for the LysM1–LysM3 groove in the HEK293-produced LysM2 mutant when compared with the HEK293-produced wild-type Ecp6 may be attributed to disabled cooperativity that may occur between both binding sites in the wild-type protein.

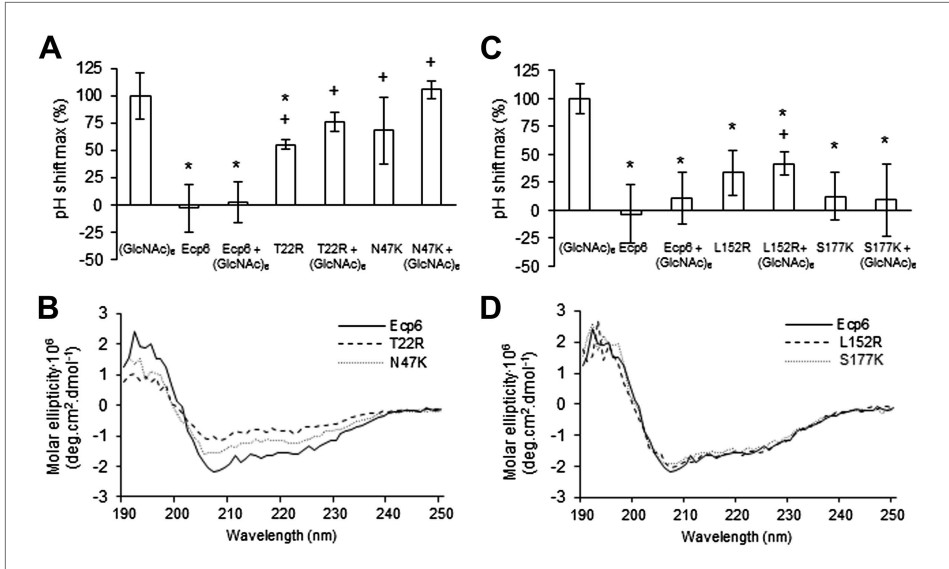

**Figure 4**. Analysis of the capacity to prevent chitin recognition by Ecp6 mutants on the chitin-binding site of LysM1 (A–B) or LysM3 (C–D). (**A** and **C**) The maximum pH shift determined on treatment with 10 nM $(GlcNAc)_6$ and 100 nM wild-type or mutants in LysM1 (**A**) and LysM3 (**B**) after normalization to treatment with $(GlcNAc)_6$ in the absence of Ecp6 protein is represented. Bars present averages of at least two replications with standard deviations. Significant differences with $(GlcNAc)_6$ treatment are indicated with asterisks and significant differences with Ecp6 treatment are indicated with plusses (*t*-test p≤0.05). (**B** and **D**) Circular dichroism spectra of the mutants in LysM1 (**B**) and LysM3 (**D**) at 10 µM.

We subsequently analyzed the role of the LysM1–LysM3 groove in suppression of chitin-triggered immunity in the tomato cell suspension assay with the HEK293-produced LysM2 mutant T95R. As expected, this mutant protein was able to prevent chitin-triggered medium alkalinization (*Figure 6*). Consequently, the composite LysM1–LysM3 binding site provides a single binding event with ultra-high (pM) affinity for chitin binding, the highest chitin-binding affinity described in nature, which is extremely competent to sequester chitin oligosaccharides. Through this effector activity, *C. fulvum* prevents chitin oligosaccharides from activating host immune receptors during infection of tomato.

## Discussion

Various fungal pathogens secrete LysM effectors to scavenge chitin fragments and thus avoid recognition by host immune receptors that activate immune responses (*Bolton et al., 2008*; *de Jonge et al., 2010*; *Marshall et al., 2011*; *Mentlak et al., 2012*; *Kombrink et al., 2011*). However, the mechanism used by these LysM effectors to efficiently compete for chitin binding with immune receptors remained unclear. Our structural and biochemical analysis of the LysM effector Ecp6 has unveiled a novel mechanism for chitin binding that evolved in fungi, in which the concerted action of two LysM domains results in sequestration of a chitin oligomer with ultra-high affinity. The flexible loop between LysM1 and the rigid LysM2–LysM3 body enables the sandwiching of four GlcNAc units of a chitin oligomer in the interface of the LysM1–LysM3 dimer by 12 hydrogen bonds and many other noncovalent interactions, resulting in ultra-high (pM) affinity. Based on the structure of the binding groove, it is anticipated that the intrachain LysM dimerization is substrate induced rather than pre-formed, as the chitin oligomer is completely buried in between the two LysM domains that are in very close proximity to make contact with the chitin oligomer. A pre-formed binding site would require a chitin oligomer to slide along the two LysM domain binding sites, which is practically impossible. At minimum, the binding groove has to breathe to let the oligomer enter before the chitin oligomer is bound. Furthermore, the noncovalent interactions among proximate residues on the complementary surfaces of LysM1 and LysM3 are likely not sufficient to keep a pre-formed binding groove in a closed conformation. More realistically, and considering the long and flexible linker that separates LysM1 from the LysM2–LysM3 body, the binding groove significantly opens in absence of the ligand

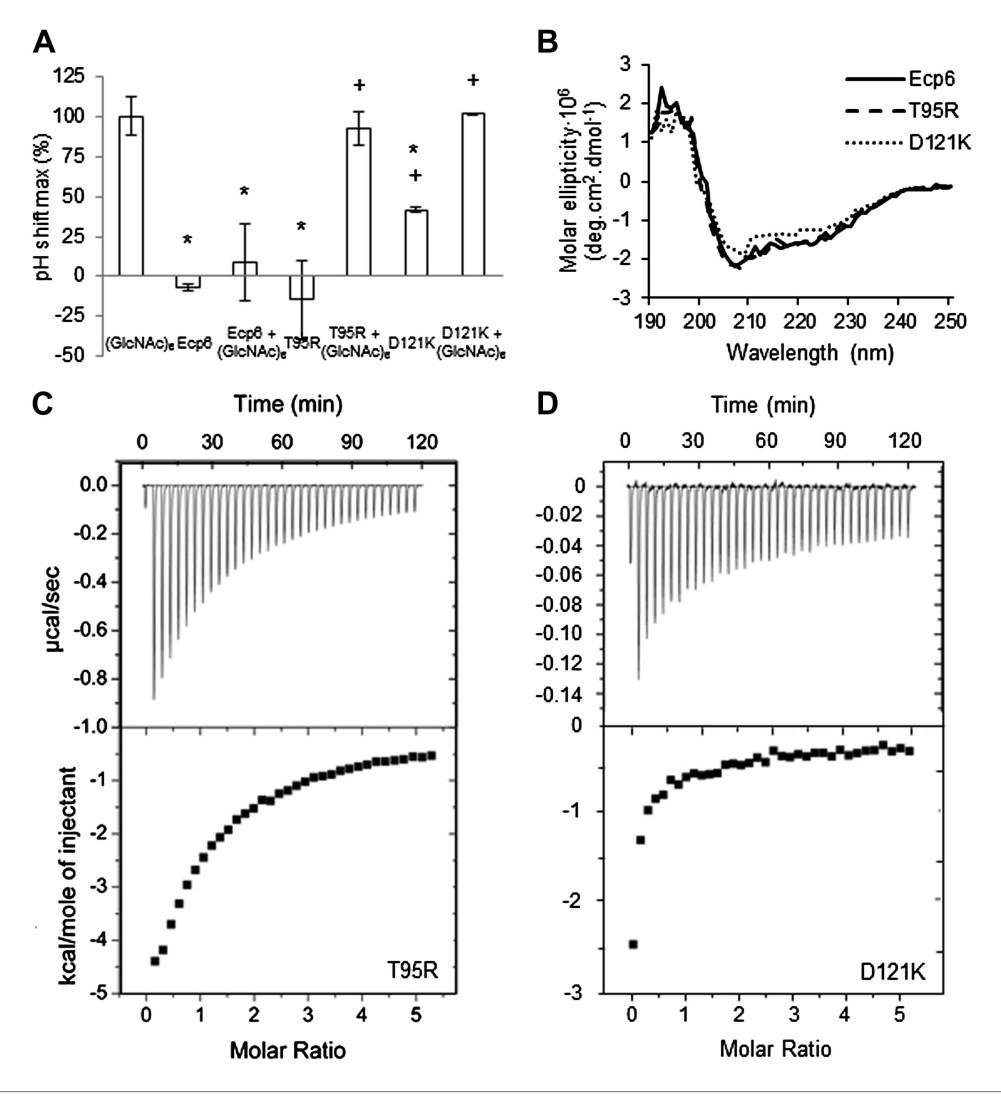

**Figure 5**. LysM2 mutants are impaired in chitin scavenging. (**A**) Prevention of chitin-triggered medium alkalinization in the tomato cell suspension assay by Ecp6 mutants. The maximum pH shift determined on treatment with 10 nM $(GlcNAc)_6$ and 100 nM Ecp6 or the mutants on LysM2 (T95R and D121K) after normalization to treatment with $(GlcNAc)_6$ in the absence of Ecp6 is represented. Bars present averages of three replicates with standard deviations. Significant differences with $(GlcNAc)_6$ treatment are indicated with asterisks, and significant differences with Ecp6 treatment are indicated with plusses (t-test $p \leq 0.05$). (**B**) Circular dichroism spectra of the mutants on LysM2 at 10 µM. (**C**) T95R and (**D**) D121K mutants on LysM2 are impaired in chitin binding. Raw data (upper panels) and integrated heat measurements (lower panels) from isothermal titration calorimetry of $(GlcNAc)_6$ binding to T95R and D121K mutants produced in *P. pastoris*.

and closes upon ligand binding, leading to substrate-induced intrachain LysM domain dimerization. Longer chitin oligomers may protrude from the Ecp6 protein into the solvent on both ends, which is particularly relevant since these are considered to be the most biologically active in eliciting chitin-triggered immune responses (*Felix et al., 1993*; *Liu et al., 2012*). Indeed, previously determined affinity constants were similar for chitin tetra-, penta-, hexa- and octamers (*de Jonge et al., 2010*). Likely, as soon as Ecp6 is secreted from the plasma membrane of *C. fulvum*, it will sequester a chitin oligo-saccharide that remains firmly attached to the protein. The global arrangement of the three LysMs of the *Arabidopsis* AtCERK1 chitin receptor, all three facing in outward direction on the surface of the protein, clearly prevents the establishment of intrachain LysM dimerization in a similar fashion as observed in Ecp6 (*Figure 8*). Moreover, only a single of the three AtCERK1 LysM domains is apparently involved

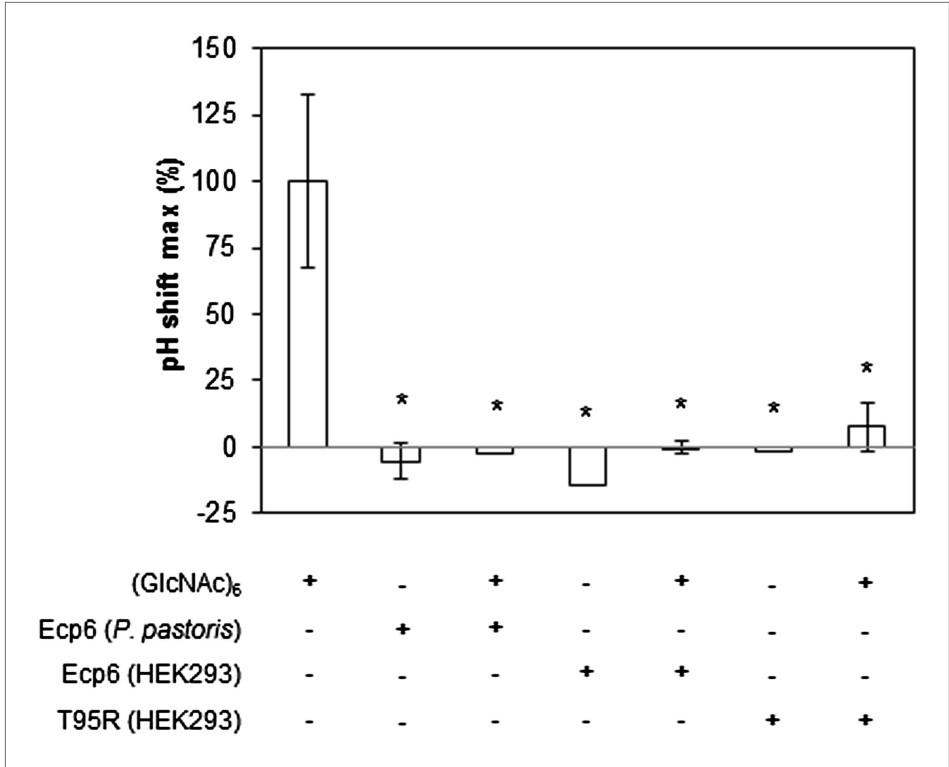

**Figure 6**. Prevention of chitin-triggered medium alkalinization in the tomato cell suspension assay by Ecp6 and T95R produced in HEK293 cells. The maximum pH shift determined on treatment with 10 nM $(GlcNAc)_6$ and 100 nM Ecp6 or T95R mutant after normalization to treatment with $(GlcNAc)_6$ in the absence of Ecp6 is represented. Bars present averages of three replicates with standard deviations. Significant differences with $(GlcNAc)_6$ treatment are indicated with asterisks, and significant differences with Ecp6 treatment are indicated with plusses ($t$-test $p \leq 0.05$).

in substrate binding, and only a single chitin oligomer is bound by a receptor molecule (**Liu et al., 2012**). Consequently, AtCERK1 binds $(GlcNAc)_6$ with a considerably lower affinity ($k_d = 44.8$ μM; **Liu et al., 2012**) than the LysM1–LysM3 binding groove of Ecp6 ($k_d = 280$ pM; **Figure 7B**). A tomato receptor for chitin has not been identified until now. However, it has been described based on radiolabeling assays that tomato cells and microsomal membranes can bind chitin oligomers with binding constants of 1.4 nM and 23 nM, respectively (**Baureithel et al., 1994**). Also, these affinities are lower than that of the LysM1–LysM3 binding groove of Ecp6. Considering also the high amount of Ecp6 effector protein that is secreted, especially during the initial stages of the infection (**Bolton et al., 2008**), these data collectively explain how *C. fulvum*, and possibly also other fungal pathogens (**de Jonge and Thomma, 2009**; **Marshall et al., 2011**; **Mentlak et al., 2012**), manage to suppress chitin-triggered immunity during infection of their hosts.

In *Arabidopsis*, AtCERK1 receptors have been suggested to bind in tandem to long chitin oligomers, prompting dimerization and activation of immune signaling (**Liu et al., 2012**). Thus, considering LysM dimerization as a mechanism for substrate binding, also interchain LysM dimerization may be exploited by nature. In this study as well as in previous studies (**de Jonge et al., 2010**), it was found that *P. pastoris*-produced Ecp6 is able to suppress chitin-triggered immunity of tomato cells although only LysM2 is available for chitin binding. Moreover, *P. pastoris*-produced LysM2 mutants in Ecp6 were impaired in prevention of chitin-triggered alkalinization of tomato cell suspensions (**Figure 5**). In this respect it is surprising that the affinity that was determined for chitin binding by LysM2 is lower than previously determined affinities for chitin binding by tomato cells (**Baureithel et al., 1994**). However, it needs to be noted that different methods have been used to determine these affinities, and that the binding assays were performed under different conditions that furthermore differ from those in planta during the interaction with the pathogen. Because the concentrations of *P. pastoris*-produced Ecp6 (100 nM) and chitin (10 nM) that were used in the alkalinization experiment are much lower than the LysM2

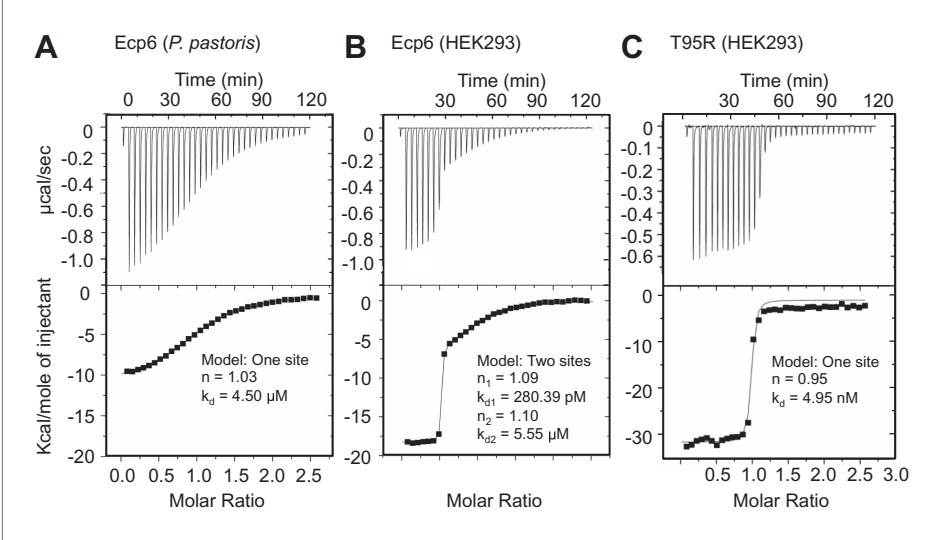

**Figure 7**. Ultra-high affinity chitin binding by intrachain LysM dimerization in Ecp6. Raw data (upper panels) and integrated heat measurements (lower panels) from isothermal titration calorimetry of $(GlcNAc)_6$ binding to Ecp6 produced in *P. pastoris* (**A**) and in HEK293 (**B**) and T95R mutant produced in HEK293 (**C**). Lines in the lower panel represent best-fit curves for one (*P. pastoris*-produced T95R) or two (HEK293-produced) binding site model.

dissociation constant (4.5 µM), *P. pastoris*-produced Ecp6 will only sequester a small portion of the available chitin oligosaccharides, and the amount of remaining, unbound, chitin oligosaccharides should be sufficient to activate chitin-triggered immunity. Thus, the observation that LysM2 is able to suppress chitin-triggered immunity strongly suggests that this LysM suppresses chitin-triggered immunity through another mechanism than through chitin oligosaccharide sequestration. Potentially, LysM2 may be involved in perturbation of the activation of chitin-triggered immunity by preventing the immune receptor dimerization that is required for the activation of immune signaling (*Liu et al., 2012*). Ecp6 may bind to chitin oligomers that have been bound by host immune receptor monomers, thus physically blocking

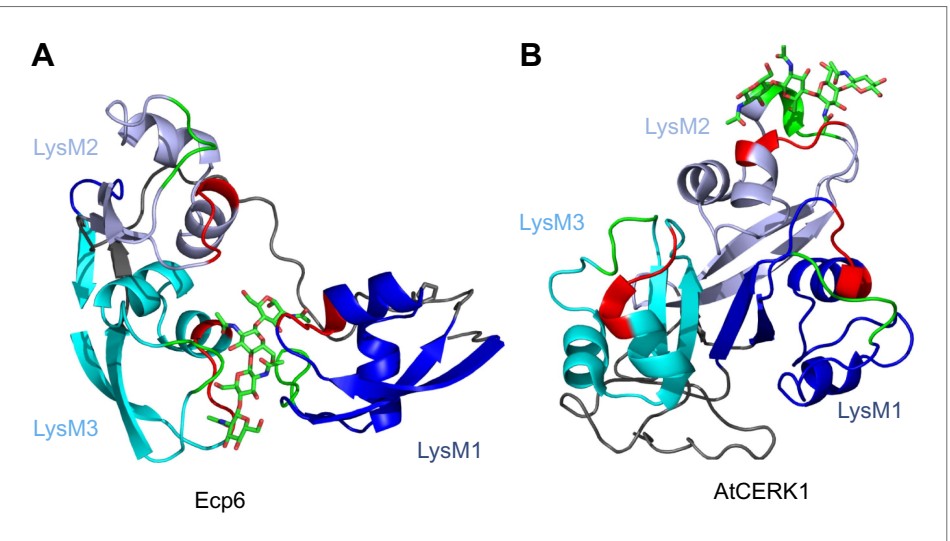

**Figure 8**. Spatial distribution of the LysM domains in the *Arabidopsis thaliana* chitin-binding immune receptor AtCERK1. In contrast to Ecp6 (**A**), the global arrangement of the LysMs (colored in three shades of blue and with chitin-binding loops in red and green) in the AtCERK1 ectodomain does not allow the formation of an intrachain LysM dimer (**B**). Only one of the three LysMs is reported to bind a chitin oligomer (*Liu et al., 2012*).

host immune receptor dimerization. These mechanisms may explain how LysM2 of Ecp6 may perturb chitin-triggered immune signaling. In conclusion, it appears that the *C. fulvum* LysM effector Ecp6 is able to suppress chitin-triggered immunity through two mechanisms; efficient chitin oligosaccharide sequestration through a ligand-induced composite LysM1–LysM3 binding groove, and secondly a mechanism exerted by LysM2 that does not involve chitin oligosaccharide sequestration but may involve perturbation of host immune receptor complexes. Future studies should reveal whether, and how, perturbation of chitin-triggered immunity by LysM2 occurs in the interaction of *C. fulvum* with tomato.

Convergent evolution towards recognition of the same ligand by molecules of taxonomically diverse organisms frequently occurs, as exemplified by the perception of bacterial flagellin by the mammalian and plant immune receptors TLR5 and FLS2, although these receptors recognize different PAMP motifs within the same ligand (*Felix et al., 1999*; *Smith et al., 2003*; *Chinchilla et al., 2006*). In contrast, our study on chitin perception by LysM proteins shows that convergent evolution has brought about different modes to interact with the same PAMP motif in taxonomically diverse organisms by structurally diverse proteins (*Figure 8*).

## Materials and methods

### Protein production

The pGEM-T (Invitrogen, Carlsbad, CA) vector containing $His_6$-FLAG tagged Ecp6 (*de Jonge et al., 2010*) was used to obtain the mutant constructs by PCR using overlapping primers (T22R: forward TGACCGCCTC ACCTCCATTG and reverse TGAGGCGGTCACCCTTGAC; N47K: forward CCCCAAACTCATCGAG CTCGGCGC and reverse CGATGAGTTTGGGGTTGGCGAG; T95R: forward GAACCCAAAGGCCAT CGATGTTGG and reverse GTGAGACGGTCGCCGCTG; D121K: forward GAACCCAAAGGCCATCG ATGTTGG and reverse CGATGGCCTTTGGGTTCTCGATCTG; L152R: forward GTGACCGTTTCGTCG ATTTGG and reverse ACGAAACGGTCACCGGCC; S177K: AACCCAAAGAAGCTCAAGGTTGGTCAGC and reverse GAGCTTCTTTGGGTTAACGTTGTTG) that contained the corresponding nucleotide mismatch, followed by digestion of the template DNA by *DpnI*. A pPIC9 vector (Invitrogen) containing $His_6$-FLAG tagged Ecp6 was used to express Ecp6 in *Pichia pastoris* (*de Jonge et al., 2010*). Purification was performed using a $Ni^{2+}$-NTA Superflow column (Qiagen, Valencia, CA). Chitin-free Ecp6 was produced deploying HEK293 cells (Genscript, Piscataway, NJ). After codon optimization, a signal peptide sequence in *Ecp6* gene construct, which already contained a $His_6$ and Flag tag on the N-terminus, was added. The recombinant plasmids encoding Ecp6 and the T95R mutant were transiently transfected into 100 ml suspension of HEK293 cell cultures. The target protein was captured from the cell culture supernatant by HiTrap chelating HP 5 ml (GE Healthcare, Milwaukee, WI) followed by buffer exchange. The purified proteins were analyzed using SDS-PAGE and Western blot using the primary antibody Mouse-anti-His mAb. Quantification was performed based on the absorbance of the protein solution at 280 nm.

### Chitin-triggered alkalinization of tomato cell suspension

Medium alkalinization experiments were performed as previously described (*Felix et al., 1993*; *de Jonge et al., 2010*). Briefly, 2.5 ml aliquots of a suspension of tomato cell line Msk8 in 12-well culture plates on a rotary shaker at 200 rpm were allowed to equilibrate for at least 2 hr. On addition of chitin oligosaccharides (Isosep AB, Tullinge, Sweden), the pH of the medium was measured, and the maximum increase of the pH ($\Delta pH_{max}$) that occurred within 3–5 min after application of chitin oligosaccharides was calculated. As previously noted by others as well, the maximum pH shift obtained after chitin stimulation varied little within an experiment when using the same batch of cells, but varied significantly between different experiments when using different batches of cells (*Felix et al., 1998*). Consequently, the maximum pH shift varied between 0.06 and 0.14 for the different experiments. In each experiment, the $\Delta pH_{max}$ was normalized to a $(GlcNAc)_6$ control (10 nM). Prior to addition, mixtures of Ecp6 protein (100 nM) and chitin oligosaccharides (10 nM) were kept at room temperature for at least 10 min while shaking gently. All experiments were performed at least three times.

### Crystallization conditions and structure determination

*P. pastoris*-produced Ecp6 was further purified by gel filtration chromatography (Superdex 75; GE Healthcare) in 20 mM HEPES, pH 7.0, and 50 mM NaCl. Large single crystals were only obtained by micro-seeding. To this end, intertwined plate-like crystals that grew overnight in the initial Ecp6 vapor-diffusion crystallization screening were harvested, smashed in stabilizing mother liquid, and

used for micro-seeding (*Bergfors, 2003*), using a reservoir with 200 mM ammonium sulfate, 100 mM sodium acetate, pH 4.6, and 20–30% PEG MME 2000 (wt/vol). A SAD experimental dataset was obtained from an I3C (5-amino-2,4,6-triiodoisophthalic acid; Jena Biosciences, Jena, Germany) soaked Ecp6 crystal at ESRF beamline ID29 (Grenoble, France) (*Gabadinho et al., 2010*). A native high-resolution dataset was collected at BL14.1 of the BESSY II storage ring (Berlin, Germany) (*Mueller et al., 2012*). Datasets were processed with MOSFLM (*Leslie and Powell, 2007*) and XDS (*Kabsch, 2010*) and scaled and merged using SCALA (*Evans, 1997*). Initial phases were calculated using PHENIX AUTOSOL (*Adams et al., 2002*). Automatic model building failed because of the sequence homology between the individual LysM domains and because of the additional density for the ligand. Therefore, manual model building in COOT (*Emsley and Cowtan, 2004*) was used instead. The structure model was finally refined to 1.6 Å resolution using REFMAC5 (*Vagin et al., 2004*). The structure was validated using the wide range of tools offered by the program COOT. In addition, structures were validated by the RCSB Protein Data Bank (PDB) services as part of the Auto Deposition Input Tool (ADIT) process (*Berman et al., 2000*). All structural figures were created with PyMOL (DeLano Scientific/Schroedinger).

## Isothermal titration calorimetry

Isothermal titration calorimetry (ITC) experiments were performed at 25°C following standard procedures using a Microcal VP-ITC calorimeter (GE-Healthcare). *P. pastoris*-produced wild-type Ecp6 (30 µM) and the mutants L152R (38 µM), S177K (30 µM), T22R (30 µM), N47K (20 µM), T95R (30 µM), and D121K (20 µM), containing a chitin tetramer in the LysM1-LysM3 groove, or 8 µM of chitin-free Ecp6 or T95R mutant, produced in HEK293 cells, were titrated with 1 injection of 1 µl, followed by 33 injections of 8 µl of $(GlcNAc)_6$ (Isosep AB, Tullinge, Sweden) at 400 µM for *P. pastoris*-produced Ecp6, S177K, T22R, and N47K, 200 µM for *P. pastoris*-produced L152R, 800 µM for *P. pastoris*-produced T95R, or 200 µM for HEK293 produced-Ecp6. Both ligand and protein were suspended in PBS, pH 7.2. Data were analyzed using Origin (OriginLab) and fitted to the models describing one (for *P. pastoris*-produced protein) and two types (for HEK293-produced Ecp6) of binding sites. Experiments were repeated three times with similar results.

## Circular dichroism

CD spectra were recorded on Jasco J-715 from 190 to 250 nm at 24°C using a 0.1 cm path length cell. Proteins were at a final concentration of 10.6 µM in water. Measurements were recorded at 1 nm wavelength increments at 100 nm/min by using a 1 nm bandwidth, 0.25 s response time. Final spectra are the average of four replicates.

## Accession codes

For structural data, see Protein Data Bank ID codes 4B8V and 4B9H.

## Acknowledgements

Rolf Hilgenfeld (University of Lübeck) is gratefully acknowledged for support. JRM thanks the DFG Cluster of Excellence 'Inflammation at Interfaces' (EXC 306).

## Additional information

### Funding

| Funder | Grant reference number | Author |
| --- | --- | --- |
| Marie Curie COFUND action and the Universidad Politécnica de Madrid | | Andrea Sánchez-Vallet |
| NGI Young Visiting Scientist Stipend | | Bart PHJ Thomma |
| Netherlands Organization for Scientific Research (NWO-ALW) | | Bart PHJ Thomma |
| Centre for BioSystems Genomics (CBSG) | | Bart PHJ Thomma |
| Deutsche Forschungsgemeinschaft KFO-126 Grant | ME2741/2 | Jeroen R Mesters |

The funders had no role in study design, data collection and interpretation, or the decision to submit the work for publication.

## Author contributions

AS-V, Acquisition of data, Analysis and interpretation of data, Drafting or revising the article; RS-B, Acquisition of data, Analysis and interpretation of data; AK, D-JV, Helpful discussions and critical reflections, Acquisition of data, Contributed unpublished essential data or reagents; GH, Analysis and interpretation of data, Drafting or revising the article; BPHJT, Conception and design, Analysis and interpretation of data, Drafting or revising the article; JRM, Conception and design, Acquisition of data, Analysis and interpretation of data, Drafting or revising the article

## Additional files

### Major datasets

The following datasets were generated:

| Author(s) | Year | Dataset title | Dataset ID and/or URL | Database, license, and accessibility information |
|---|---|---|---|---|
| Saleem-Batcha R, Sanchez-Vallet A, Hansen G, Thomma BPHJ, Mesters JR | 2012 | Cladosporium fulvum LysM effector Ecp6 in complex with a beta-1,4- linked *N*-acetyl-D-glucosamine tetramer | 4B8V; http://www.rcsb. org/pdb/search/ structidSearch.do? structureId=4B8V | Publicly available at the RCSB Protein Data Bank (http://www. rcsb.org/). |
| Saleem-Batcha R, Sanchez-Vallet A, Hansen G, Thomma BPHJ, Mesters JR | 2012 | Cladosporium fulvum LysM effector Ecp6 in complex with a beta-1,4- linked *N*-acetyl-D-glucosamine tetramer: I3C heavy atom derivative | 4B9H; http://www. rcsb.org/pdb/search/ structidSearch.do? structureId=4B9H | (http://www.rcsb.org/). |

The following previously published dataset was used:

| Author(s) | Year | Dataset title | Dataset ID and/or URL | Database, license, and accessibility information |
|---|---|---|---|---|
| Chai J, Liu T, Han Z, She J, Wang J | 2012 | Chitin elicitor receptor kinase 1 | 4EBZ; http://www. rcsb.org/pdb/explore/ explore.do?structureId= 4EBZ | Publicly available at the RCSB Protein Data Bank (http://www. rcsb.org/). |

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
