## [Decision Letter]

Thank you for sending your work entitled “Fungal effector Ecp6 outcompetes host immune receptor for chitin binding through intrachain LysM dimerization” for consideration at *eLife*. Your article has been favorably evaluated by a Senior editor, Detlef Weigel, and 3 reviewers, one of whom is a member of our Board of Reviewing Editors.

The Reviewing editor and the other reviewers discussed their comments before we reached this decision, and the Reviewing editor has assembled the following comments to help you prepare a revised submission.

The authors wish to understand the molecular basis by which ECP6, a secreted fungal protein, suppresses chitin-induced responses in tomato. ECP6 has LysM domains that bind chitin. The authors propose that it is the ability of these LysM domains within ECP6 to bind chitin that is responsible for immune suppression, through a model in which chitin is directly sequestered from the receptor.

The current work employs structural analysis to guide functional and biochemical experiments. The excellent structural analysis was based on protein expressed in *Pichia pastoris*. ECP6 purified from Pichia already contains GlcNAc4. It is argued that a longer GlcNAc chain could possibly bind in the same way. This is relevant since longer chains are more active as elicitors of defense.

Based on the structure and binding measurements, the authors argue that the high affinity binding site composed of LysM1-3 is essential for the ability of ECP6 to suppress chitin-induced innate immunity. However, the data supporting this in the tomato alkalinization assay are not presented (this could be done using the T95R mutant produced by HEK293 cells: this data is alluded to in the legend of Figure 6, but the figure itself does not show this). Given that the LysM1–3 site is blocked either by NAG4 or mutation in all other alkalinization assays, it is only mutations in the low affinity LysM2 that are shown to affect the ability to suppress innate immunity.

A major concern for the authors to address is how a low affinity interaction can effectively suppress chitin-induced alkalinization. Chitin is a highly potent elicitor; some biological effects can be measured with nmol or even pmol amounts of chitin. Judging from the amount of chitin used by the authors and the amount of ECP6, it appears that a lot of unbound chitin would still be available to elicit alkalinization. In the absence of Pichia-produced Ecp6, 10 nM of chitin (NAG6) elicits the alkalization response. With Pichia-produced Ecp6, the concentration of free chitin should still be 9.8 nM. It seems most unlikely that tomato cells are only sensitive to 10 nM chitin, but not to 9.8 nM. A dose response experiment with chitin should be done to establish the sensitivity of the assay, preferably also with Ecp6 produced by yeast and by mammalian cells. If you are near the threshold of responsiveness for this assay, it might explain how a slight reduction in the total chitin could result in reduced alkalinization. Otherwise, assuming the ITC binding data are correct, the LysM2 domain does not directly out-compete for chitin-induced responses in the assay, and something else must be going on. The authors need to discuss this point and consider that their model for how ECP6 affects defense activation by chitin via the LysM2 domain may not be correct.

The authors discussed the possibility that since AtCERK1 has been reported to be a low affinity chitin receptor, the low affinity of LysM2 is sufficient to compete and this explains how ECP6 can suppress innate immunity. The authors should re-consider this argument in light of the wealth of data in the field showing that chitin oligomers induce a variety of plant innate immunity responses at low nM and even pM levels. Additionally, they are using the tomato bioassay. In the classic experiments, chitin binds to tomato cells with very high affinity. Hence, as mentioned above, the uM levels of chitin binding by Ecp6 LysM2 do not seem to be sufficient to explain suppression of alkalinization.

Additional points that the authors should address are outlined below:

1) The results of the tomato protoplast assay are an important part of the manuscript. They should be mentioned in the Abstract (and maybe at the end of the Introduction as well), especially in the context of the structure-based mutants that no longer support suppression of chitin-triggered immunity. This would directly link the in vitro and in vivo assays.

2) The authors state that the L152R and S177K mutants are not autoactive, but they do show some pH shift relative to Ecp6. The authors need to provide statistical analysis of all the alkalinization data. The legend should include what the maximum pH shift was for each assay (and as mentioned above, a dose response establishing the sensitivity of the assay should be provided).

3) The authors favor a model in which intrachain dimerization is substrate induced. What is the evidence for this: why shouldn't the binding site be pre-formed in the protein?

4) Do you actually know that the chitin oligomer bound to Ecp6 purified from Pichia is (GlcNAc)4? Could it be longer? It could be possible to positively identify this by mass spectrometry.

5) Please further discuss or modify the presentation of the ITC in Figure 3. Were the T22R and N47K data fitted to a model (indicated by the lines joining the points)? If so, what was the result?

6) The authors need to comment on the differences obtained from their previous paper (de Jonge et al, Science 2010): in that paper, three binding events were documented; here it is two. What is the explanation for the difference?

[Editors' note: further clarifications were requested prior to acceptance, as described below.]

The reviewers agree that the publication of the ECP6 structure will be a significant contribution to the field. It is clear that the LysM1–3 binding site has sufficient affinity to work as the authors suggest. However, since the LysM2 site can also work, apparently, in the absence of LysM1–3, it does beg the question as to what mechanism, either direct competition or, perhaps, other events, are the major factor in the action of ECP6.

The reviewers previously brought up the problem of interpretation of the chitin binding assays. Specifically, calculation of the equilibrium between protein and chitin levels indicates that significant free chitin would exist in their assays. At the concentrations being used, well below the measured K_d_ for the LysM1–3 mutant (with LysM2 intact), a 10-fold excess of protein over ligand will not result in all the ligand being bound to the protein. In fact, virtually none of it will be sequestered. Therefore, prior to adding the pre-incubation mix to the tomato cells, 0.22 nM complex is formed leaving 99.8 nM free protein and 9.8 nM free ligand. If the authors can provide an alternative mathematical explanation for how the protein can sequester the ligand under the conditions used, they should show this explicitly.

One hypothesis that could lead to the sequestration of chitin in solution is the presence of an “extra factor” in the assay mix that significantly increases the affinity of Ecp6-LysM2 for chitin (lowering the K_d_). It would be pure speculation what this could be. The authors do indicate in the manuscript that the LysM2 binding domain is unlikely to work via sequestering chitin, but they should make this conclusion stronger given the calculation above.

The authors claim to satisfy very different binding models to their ITC data for Ecp6 (comparison of past published work and present analysis). Therefore, the authors should provide some control ITC traces looking at buffer injections into the protein, ligand injection into buffer (no protein) and buffer to buffer. This will demonstrate that the heats measured are coming from the interactions. It is satisfactory that this is only provided for wild-type Ecp6.

---

## [Author Response]

*Based on the structure and binding measurements, the authors argue that the high affinity binding site composed of LysM1-3 is essential for the ability of ECP6 to suppress chitin-induced innate immunity. However, the data supporting this in the tomato alkalinization assay are not presented (this could be done using the T95R mutant produced by HEK293 cells: this data is alluded to in the legend of Figure 6, but the figure itself does not show this). Given that the LysM1–3 site is blocked either by NAG4 or mutation in all other alkalinization assays, it is only mutations in the low affinity LysM2 that are shown to affect the ability to suppress innate immunity*.

The ultra-high affinity binding of chitin by the composite LysM1–LysM3 site is the highest affinity reported in nature, which already indicates that the binding site outcompetes the plant receptor in order to prevent chitin triggered immunity. To provide further evidence, we have now included the results of assays using LysM2 mutant T95R that was produced in mammalian cells in the tomato cell suspension assay. As expected, we found that this mutant prevents activation of chitin-triggered immunity in this tomato cell suspension, which is a direct demonstration of the relevance of the ultrahigh affinity binding groove to outcompete plant chitin receptors. These data are now added to Figure 6.

*A major concern for the authors to address is how a low affinity interaction can effectively suppress chitin-induced alkalinization. Chitin is a highly potent elicitor; some biological effects can be measured with nmol or even pmol amounts of chitin. Judging from the amount of chitin used by the authors and the amount of ECP6, it appears that a lot of unbound chitin would still be available to elicit alkalinization. In the absence of Pichia-produced Ecp6, 10 nM of chitin (NAG6) elicits the alkalization response. With Pichia-produced Ecp6, the concentration of free chitin should still be 9.8 nM. It seems most unlikely that tomato cells are only sensitive to 10 nM chitin, but not to 9.8 nM. A dose response experiment with chitin should be done to establish the sensitivity of the assay, preferably also with Ecp6 produced by yeast and by mammalian cells. If you are near the threshold of responsiveness for this assay, it might explain how a slight reduction in the total chitin could result in reduced alkalinization. Otherwise, assuming the ITC binding data are correct, the LysM2 domain does not directly out-compete for chitin-induced responses in the assay, and something else must be going on. The authors need to discuss this point and consider that their model for how ECP6 affects defense activation by chitin via the LysM2 domain may not be correct*.

*The authors discussed the possibility that since AtCERK1 has been reported to be a low affinity chitin receptor, the low affinity of LysM2 is sufficient to compete and this explains how ECP6 can suppress innate immunity. The authors should re-consider this argument in light of the wealth of data in the field showing that chitin oligomers induce a variety of plant innate immunity responses at low nM and even pM levels. Additionally, they are using the tomato bioassay. In the classic experiments, chitin binds to tomato cells with very high affinity. Hence, as mentioned above, the uM levels of chitin binding by Ecp6 LysM2 do not seem to be sufficient to explain suppression of alkalinization*.

We respectfully disagree with the statement that a lot of unbound chitin is still available in our cell suspension assays to elicit medium alkalinization. In these assays, we use 10 nM of chitin and 100 nM (a 10-fold excess) of Ecp6 protein. As chitin binds to LysM2 (the only site available in *Pichia pastoris*-produced Ecp6) with 1:1 stoichiometry, all the chitin that is added to the tomato cell suspension will be bound by LysM2, and no chitin will be released to the medium. In addition, it has previously been described that Ecp6 produced in *P. pastoris* is able to prevent chitin-triggered immunity even when administered in equimolar concentrations with Ecp6 (de Jonge et al., Science 2010), which is again in agreement with the 1:1 stoichiometry. Furthermore, in the same manuscript (de Jonge et al., Science 2010) we have shown that 10 times less chitin (1 nM) is still able to induce a pH shift which is similar to that triggered by 10 nM of chitin. In conclusion, in our assays we intentionally use a ten-fold excess of Ecp6 protein in order to avoid being near the threshold of responsiveness.

An issue that indeed remains concerns potential discrepancies of affinities of LysM2 of Ecp6 and plant chitin receptors. However, it needs to be noted that different affinities have been reported based on completely different types of binding assays, and it cannot be excluded that these assays contribute to the observed differences in affinities. Affinities of tomato cells for chitin binding have been determined with radiolabeling (Baureithel et al., JBC 1994), whereas we have used isothermal titration calorimetry (ITC). Moreover, these different affinity measurement technologies require that binding assays are performed under specific conditions that differ from the conditions that are used for the competition assays, and also from conditions during a true plant–pathogen interaction. This may obscure the precise determination of relative affinities under the relevant conditions (i.e., during competition in planta). Thus, very precise comparisons of such affinities should be taken with care. For this reason, we have compared the affinity constants of AtCERK1 (Liu et al., Science 2012) and Ecp6 as these were both determined by the same technology, ITC, and under very similar conditions. Finally, previously performed competition assays involving *P. pastoris*-produced Ecp6 and rice microsomal membranes showed that Ecp6 can compete for chitin binding with plant receptors (de Jonge et al., Science 2010). Nevertheless, we agree with the reviewers that Ecp6 may still affect defense activation by chitin via the LysM2 domain in other ways than through sequestration. Therefore, at the end of the Discussion it is stated: “Alternatively, LysM2 may be involved in perturbation of the activation of chitin-triggered immunity by preventing the immune receptor dimerization that is required for the activation of immune signaling activation (30). Ecp6 may bind to chitin oligomers that have been bound by host immune receptor monomers, thus physically blocking host immune receptor dimerization. These mechanisms may explain how LysM2 of Ecp6 may perturb chitin-triggered immune signaling.”

*1) The results of the tomato protoplast assay are an important part of the manuscript. They should be mentioned in the Abstract (and maybe at the end of the Introduction as well), especially in the context of the structure-based mutants that no longer support suppression of chitin-triggered immunity. This would directly link the* in vitro *and* in vivo *assays*.

This change has been made as requested.

*2) The authors state that the L152R and S177K mutants are not autoactive, but they do show some pH shift relative to Ecp6. The authors need to provide statistical analysis of all the alkalinization data. The legend should include what the maximum pH shift was for each assay (and as mentioned above, a dose response establishing the sensitivity of the assay should be provided)*.

T-test analyses have been now performed on all the alkalinization experiments. In addition, we have included data from additional assays to have even more data points allowing a more robust statistical analysis. As indicated in Figure 4, the differences in the pH shift by Ecp6 wild-type protein and by L152R of S177K are not significant. Consequently, we do not consider these proteins to be autoactive.

As previously described, the maximum pH shift obtained after stimulation varied little within one experiment when using one batch of cells, but varied significantly between different experiments when using different batches of cells (15), explaining why we also obtained rather large differences in the maximum pH shift in our experiments. We have added information on the maximum pH shift in the Materials and methods section of the manuscript. As indicated, we previously (20) reported on a dose response experiment with Ecp6 produced in *P. pastoris*.

*3) The authors favor a model in which intrachain dimerization is substrate induced. What is the evidence for this: why shouldn't the binding site be pre-formed in the protein*?

The statement that the intrachain LysM domain dimerization is substrate-induced is inferred from the structure of the binding groove. In this groove, the chitin oligomer is completely buried, and the two LysM domains are in very close proximity to make contact with the chitin oligomer through many noncovalent interactions, including 12 hydrogen bonds. This binding groove cannot be preformed because a chitin oligomer GlcNAc cannot enter a preformed binding groove and “slide along” the two LysM domain binding sites. At a minimum, the binding groove has to “breathe” to let the oligomer enter before the chitin oligomer is bound. Furthermore, the noncovalent interactions among proximate residues on the complementary surfaces of LysM1 and LysM3 are likely not sufficient to keep a preformed binding groove in a closed conformation. More realistically, considering the long and flexible linker that separates LysM1 from the LysM2–LysM3 body, the binding groove opens significantly in absence of the ligand, and closes upon ligand binding; hence substrate-induced intrachain LysM domain dimerization. This is now also clarified in the Discussion section.

*4) Do you actually know that the chitin oligomer bound to Ecp6 purified from Pichia is (GlcNAc)4? Could it be longer? It could be possible to positively identify this by mass spectrometry*.

We do not see any evidence for a longer chitin chain in the structure, so we are confident that only tetramers are found in the structure. It is highly unlikely that only chitin tetramers are available in *P. pastoris* cultures, but according to the structure only four units of a longer chitin oligomer are in contact with the binding groove. Most likely, these four units are protected by the binding groove during all the procedures related to fermentation, purification and crystallization of the protein, and we anticipate that protruding ends have been damaged or removed.

*5) Please further discuss or modify the presentation of the ITC in Figure 3. Were the T22R and N47K data fitted to a model (indicated by the lines joining the points)? If so, what was the result*?

The ITC curve of LysM1 mutant T22R could be fitted to a “one binding site” model and that of N47K could be fitted to a “two binding site” model, respectively. Mutant T22R binds chitin with similar thermodynamic values as Ecp6 produced in *P. pastoris* (n = 0.75; k_d_ = 4.69 µM), suggesting that we are measuring binding by LysM2. Remarkably, the second binding event of N47K revealed similar biochemical parameters (n = 0.984; k_d_ = 5.2 µM), revealing that this mutant has two LysM2-like binding sites. These results likely reflect chitin binding to LysM2 and to the partially disrupted LysM1-LysM3 groove, as only one of these two LysMs was mutagenized in a single mutant. These data are now added to the Results section of the manuscript.

*6) The authors need to comment on the differences obtained from their previous paper (de Jonge et al, Science 2010): in that paper, three binding events were documented; here it is two. What is the explanation for the difference*?

ITC measurements always have to be fit to mathematical models in order to be able to determine binding affinities. Based on several reports showing only 1:1 stochiometry for LysM domains in the past, we performed ITC measurements and were able to fit the data to a “one binding site” model, revealing three binding sites for these oligosaccharides per Ecp6 molecule. Based on the crystal structure generated in our present work, and the observation of the composite chitin binding groove, we now know that Ecp6 produced in *P. pastoris* has only one binding site available. Thus, in hindsight with the knowledge of today, the model that was used in our previous study was not correct. We have now explained this discrepancy in the manuscript.

[Editors’ note: further clarifications were requested prior to acceptance, as described below.]

*The reviewers previously brought up the problem of interpretation of the chitin binding assays. Specifically, calculation of the equilibrium between protein and chitin levels indicates that significant free chitin would exist in their assays. At the concentrations being used, well below the measured K*_*d*_
*for the LysM1*–*3 mutant (with LysM2 intact), a 10-fold excess of protein over ligand will not result in all the ligand being bound to the protein. In fact, virtually none of it will be sequestered. Therefore, prior to adding the pre-incubation mix to the tomato cells, 0.22nM complex is formed leaving 99.8 nM free protein and 9.8 nM free ligand. If the authors can provide an alternative mathematical explanation for how the protein can sequester the ligand under the conditions used, they should show this explicitly*.

We apologize for the misunderstanding; we missed the point that was made previously. Indeed, the referees are correct when stating that, at the concentrations that were used being below the K_d_ of the LysM2, sufficient chitin oligosaccharide should be available to elicit a response in the tomato cell suspension. As it is evident that LysM2 can prevent the activation of chitin-triggered immunity, it is very likely that this perturbation is established through another mechanism than merely chitin oligosaccharide sequestration by LysM2. Whereas we stated previously that “… we agree with the reviewers that Ecp6 may still affect defense activation by chitin via the LysM2 domain in other ways than through sequestration”, we agree that this statement is not correct. It is actually very unlikely that LysM2 functions through chitin sequestration, making that Ecp6 can interfere with chitin-triggered immunity in two different ways; sequestration by the LysM1–3 groove and another, complementary, mechanism that does not involved sequestration by LysM2.

*One hypothesis that could lead to the sequestration of chitin in solution is the presence of an “extra factor” in the assay mix that significantly increases the affinity of Ecp6-LysM2 for chitin (lowering the K*_*d*_*). It would be pure speculation what this could be. The authors do indicate in the manuscript that the LysM2 binding domain is unlikely to work via sequestering chitin, but they should make this conclusion stronger*.

An extra factor may be unlikely, as this extra factor in our assays cannot be *Cladosporium fulvum*-derived, the pathogen from which Ecp6 was isolated. As we used Ecp6 produced by the non-pathogenic yeast *Pichia pastoris* and mammalian Hek293 that was purified with affinity purification, a potential extra factor in the tomato cell suspension assays should almost be plant-derived. Anyway, as stated above, we agree with the referees that LysM2 is unlikely to work via sequestering of chitin and we have now stated much more explicitly that LysM2 is unlikely to function through chitin sequestration and that the two chitin binding sites of Ecp6, the LysM1–3 groove and LysM2, must function through different mechanisms.

*The authors claim to satisfy very different binding models to their ITC data for Ecp6 (comparison of past published work and present analysis). Therefore, the authors should provide some control ITC traces looking at buffer injections into the protein, ligand injection into buffer (no protein) and buffer to buffer. This will demonstrate that the heats measured are coming from the interactions. It is satisfactory that this is only provided for wild-type Ecp6*.

ITC control experiments involving chitin ligand into the buffer (PBS), of buffer injection into the buffer, and buffer injection into Ecp6 protein solution, are included in panel A of Figure 3. No heat of binding is observed in any of these control injections, confirming that the heats that were measured in the other interactions are indeed coming from the protein–ligand interactions.